# Taking Up and Terminating Leisure-Time Physical Activity over the Life Course: The Role of Life Events in the Familial and Occupational Life Domains

**DOI:** 10.3390/ijerph18189809

**Published:** 2021-09-17

**Authors:** Lars Lenze, Claudia Klostermann, Markus Lamprecht, Siegfried Nagel

**Affiliations:** 1School of Education, University of Applied Sciences and Arts Northwestern Switzerland, 5210 Windisch, Switzerland; Claudia.klostermann@fhnw.ch; 2Institute of Sport Science, University of Bern, 3012 Bern, Switzerland; siegfried.nagel@ispw.unibe.ch; 3Lamprecht und Stamm Sozialforschung und Beratung, 8032 Zurich, Switzerland; markus.lamprecht@lssfb.ch

**Keywords:** leisure-time physical activity, life events, life course, multilevel discrete-time event-history analysis, family

## Abstract

Leisure-time physical activity (LTPA) is associated with various health-promoting effects. However, little is known about the relationship between life events and changes in LTPA over the life course, especially when multiple life events occur simultaneously. Therefore, this study examines taking up and terminating LTPA associated with life events in the familial and occupational life domains over 16 years of 16–76-year-old Swiss inhabitants (*n* = 1857) in a retrospective longitudinal cohort design, using a validated telephone survey and multilevel discrete-time event-history analyses. The results show that taking up LTPA was more likely when ending a relationship and retiring and less likely when becoming a parent; terminating LTPA was more likely when ending a job, starting vocational training after 30 years, a relationship ended for men, and becoming a mother with increasing age. If experiencing multiple life events simultaneously, the greater the number of life events, the more likely persons aged 45–70 years were to take up LTPA and, conversely, the more likely persons aged 15–44 years to terminate LTPA. The relationship between life events and changes in LTPA over the life course was often age dependent, especially when experiencing multiple life events simultaneously. The findings should be considered when promoting LTPA.

## 1. Introduction

Regular leisure-time physical activity (LTPA) is a well-known behavior for health promotion [1,2], including in the long term [3]. Considering the high health costs caused by physical inactivity [4], regular and long-term LTPA is desirable from health and economic perspectives. However, research indicates that LTPA has low stability over the life course [5,6,7]. Changing life circumstances are one reason for changes in LTPA [8] and can be triggered by life events (e.g., birth of a child), which have been shown to have an effect on LTPA [9,10]. Examining life events should aid us in gaining a better understanding of the (in)stability of LTPA over the life course [11].

A significant amount of research on the relationship between certain life events and changes in different forms of physical activity has been conducted with various research designs. Typical life events investigated are changes in relationships [9,10,12], becoming a parent [9,10,12,13], changes in vocational training and occupation [9,10,14], and retirement [10,15]. These studies typically focus on only one or few of these life events separately. These life events are then associated with general physical activity or specific types thereof, such as LTPA. This information is collected via self-report or, in rare cases, objectively for general PA. Prospective and retrospective designs are used for this purpose. Life events and changes in physical activity have been studied with a focus on a single age group, from adolescents to the elderly, but especially in the context of young adulthood. The study period ranges from a few months to beyond 30 years in rare cases, with an average of about 5 years. In summary, three research gaps have been noted. First, mainly declines in or termination of LTPA have been studied rather than taking up LTPA (again). Second, often, the period of investigation focuses only on a few years in a single age group, meaning that different effects of the same life event on LTPA in relation to age or life stage have not been explored (“timing” of events [5,16]). Third, solely separate life events have been considered.

Here, we used a life-course approach and examined taking up and terminating LTPA associated with different and multiple life events in the familial and occupational life domains over 16 years for Swiss inhabitants (*n* = 1857; 16–76 years) in a retrospective longitudinal cohort design. Consequently, the picture of the relationship between life events and changes in LTPA can be supplemented in three ways. First, transitions for the exit and entry of LTPA were examined. Second, different age groups were considered over a long period (which illustrated the possible age-dependent effects of life events on LTPA). Third, in addition to specific life events, the effect of simultaneously occurring life events on LTPA was analyzed.

Our study is based on the life-course approach and its basic assumptions considering a long-term view of time and multiple life domains [17]. More specifically, the recently published “life course cube” [17], with its three axes of “time”, “level” (inner-individual, individual action, and supra-individual), and “life domain” provided the theoretical framework. Interdependencies and interactions within and across axes represent the life course, but it is not possible to test all interdependencies simultaneously [17,18]. We focused on the individual action in terms of the axis “level” and on the life domain of leisure as well as the familial and occupational life domains when regarding the axis “life domain”. Here, the axis “time” was understood as lifetime, whereby earlier experiences and the timing of events in the life course can be decisive for its consequences [19].

Looking first at the life domain of leisure, LTPA is seen as a health-enhancing behavior [20] and was understood here as alternations between episodes of activity and inactivity in the life course [21], which include exercise, sport, and unstructured recreation [22] (see Appendix B for further information). According to the life course cube, LTPA is dependent upon previous experiences therein (axis “time”), and LTPA can be influenced by life events in other life domains, which is also known as the “spillover effect” [23]. Such life events can be described as life experiences that influence the individual’s daily routine [24] and disrupt the person–environment fit that has been accustomed over time [25]. Two life domains in which life events occur with large disruptive potential for LTPA are the familial and occupational life domains [10]. After experiencing such life events, for instance, becoming a parent, adaptations are necessary [24,25], which can affect life in general [25] or other life domains [23]. For these adaptation processes, individual resources are necessary [23,26]. Here, life domains compete for resources (e.g., spending time for childcare vs. spending time for LTPA), life domains support or complement each other (e.g., gaining mental-health resources in LTPA for better performance at work), or life domains compensate respectively substitute each other’s resources (e.g., missing social exchange at work can be compensated by playing football with friends as an LTPA) [17,27,28]. As shown in these examples, individual resources seem pivotal when looking at life events and LTPA [29]. Furthermore, for physical activity, it has been stated that resources may be shifted away from physical activity after experiencing a life event [30].

With regard to the familial life domain, starting and ending a relationship as well as becoming a parent are relevant life events for LTPA [9,12]. Research has indicated a general negative relationship between becoming a parent and physical activity [13], with a higher decrease in LTPA for women [10]. Consequently, Hypothesis H1 can be formulated on this basis:

**Hypothesis** **H1.***Becoming a parent is positively associated with terminating LTPA—with a larger effect for women*.

Looking at changes in relationships, starting a relationship yields mixed effects on LTPA [10,31] ([31] only in Germany), whereas ending a relationship is related to an increase in LTPA, depending on sex [10]. Overall, current research suggests mixed and rather unclear effects of changes in relationships on LTPA. Due to the inconclusive state of research, the following research question (RQ) is asked: “To what extent are changes in relationships associated with taking up (RQ 1a) or terminating LTPA (RQ 1b)?”. In addition, the timing of life events is crucial for LTPA [5,16] but has not been investigated. For example, becoming a parent at the age of 15, 35, or 50 years has different consequences on a person’s life.

Within the occupational life domain, education-related and employment-related life events can affect LTPA [10]. Leaving high school (which is often combined with entering university) is associated with a decrease in physical activity [10,14], whereas leaving university is related to no change in physical activity [14]. Starting a new job is related to a decrease in physical activity in young adulthood [10,14], but changes in jobs have revealed inconclusive results for physical activity [10]. On the basis of the different relations of changes in jobs and vocational training on LTPA, the following research question arises: “To what extent are changes in jobs and vocational training associated with taking up (RQ 2a) or terminating LTPA (RQ 2b)?”. Retirement is clearly positively related with an increase in LTPA [15]; therefore, we can formulate the following:

**Hypothesis** **H2.***Retirement is positively associated with taking up LTPA*.

Several simultaneously occurring life events trigger greater adaptation processes and, consequently, redistribution of more resources. Only one study has investigated this issue and showed a more negative effect on physical activity in the occurrence of multiple life events in a prospective randomized control trial over two years in the United States [32]. As there is only one study on this topic to date, a research question rather than a hypothesis is stated: “To what extent is the number of simultaneously occurring life events associated with taking up (RQ 3a) or terminating LTPA (RQ 3b)?”.

## 2. Materials and Methods

The retrospective longitudinal study “Physical Activity During Life Course” is funded by the Swiss National Science Foundation and is part of the nationwide state survey “Sport Schweiz 2020” [33]. The study received approval from the Ethics Committee of the School of Education of the University of Applied Sciences and Arts Northwestern Switzerland (Windisch, Switzerland).

### 2.1. Sample

The sample consists of 1857 Swiss inhabitants aged 16–76 years. They were interviewed once via a telephone survey using the computer-assisted telephone interviewing (CATI) method in 2019. The random sample was recruited via the Federal Statistical Office (*n* = 753) and with persons from the panel of the survey institute (*n* = 1104). The mean age (*M*_age_) was 55.3 ± 19.09 years, and *n* = 1119 of the study cohort (60.3%) were women (see Table 1 for more information).

### 2.2. Instrument and Operationalization

The questionnaire was a further development of previous studies investigating LTPA during the life course [34]. Throughout the questionnaire, different aspects of current and past regular LTPA over the life course, as well as life events over the past 16 years, were asked retrospectively. The term “regular” refers to at least once a week. The unit for time was years. Hence, for each year of one’s life, the binary information of being physically active in leisure or not being physically active in leisure, taking up LTPA or terminating LTPA, respectively, was gathered. Thus, episodes of activity and inactivity were built.

With regard to measuring life events, information on familial and occupational life domains was gathered separately. Life events of 16 years (2004 to 2019) were recorded. From the familial life domain, three life events were captured: (1) starting a relationship, (2) ending a relationship, and (3) becoming a parent. In the occupational life domain, four life events were included: (1) starting vocational training (with a state-recognized certificate), (2) starting a job, (3) ending a job (excluding retirement), and (4) retirement.

### 2.3. Data Collection

A memory bias is a problem when measuring retrospective data. Four approaches were taken when attempting to keep this recall error as small as possible and gather reliable data. First, multiple cues and pathways were used to support autobiographical memory, such as hierarchical and sequential pathways [35,36]. The adapted questionnaire was tested and refined in an iterative process in a theoretical (autobiographical memory) and practical manner. For example, qualitative interviews (*n* = 9) using techniques such as the “think-aloud” method, confidence rating, and behavior coding were carried out [37]. Second, a separate study with the test–retest method was conducted to check the reliability of the final questionnaire. Therefore, *n* = 29 persons (17 (59%) women and 12 (41%) men) ranging from 23 years to 75 years of age (*M*_age_ = 51.41 ± 19.09) were interviewed twice with the final questionnaire 95.07 ± 15.34 days apart on average.

Considering different scale levels in the questionnaire, Krippendorff’s alpha [38] was used for analyses. The recommended convention for a reliable agreement of the data of α ≥0.80 [39] was fulfilled for all variables used (Appendix A). This criterion is in accordance with findings from similar investigations on reliability [34,40]. Third, the CATI method was chosen to enable trained interviewers to help specifically with problems regarding memory and comprehension. Trained interviewers participating later in the main study conducted 20 interviews to test the technical and content-related functioning of the questionnaire. Fourth, after the data collection was complete, the data were checked carefully for internal inconsistencies and discrepancies. In doing so, 168 persons were excluded. Consequently, from the initial 2025 interviewed persons, *n* = 1857 were used for analyses. All these efforts at the level of the questionnaire, the interview process, and in data cleaning were made to collect as reliable and valid data as possible.

### 2.4. Data Analyses

We obtained data from each year from 2004 to 2019 of all 1857 participants. Hence, a person-year file with *n* = 27,238 data was built. To calculate relationships between life events and taking up LTPA or terminating LTPA, a multilevel discrete-time event history analysis [41] was used. This multilevel type of event history analysis (which can also be called “survival analysis”) is equally a generalized linear mixed model (GLMM) and adjusts for the dependency in the person-year file. Hence, multiple measurements over time from the same person are not independent [42]. As usual for event history analysis, a specific “risk set” was made for each calculation [43]. Therefore, separate calculations for each life event linked with taking up or terminating LTPA were carried out. Consequently, only persons who were physically active in the respective year were “at risk” of terminating LTPA and, thus, were part of this risk set. The same approach applied inversely to physically inactive persons and taking up LTPA, as well as for life events (e.g., persons in a relationship are not at risk of starting a new relationship). Episodes of being physically active until becoming inactive could occur more than once in the 16 years recorded and vice versa (e.g., physically active in year 1–3, inactive in year 4–13, and again active in year 14–16). Thus, this was an event history analysis with possible recurrent or repeated events [41,43].

In total, five control variables that could affect LTPA in the life course were derived from the literature and included for each calculation as time-invariant or time-varying variables. The first was sex (time invariant). There are differences in the trajectory of LTPA between men and women in Switzerland [33] and so they were controlled for. The second control variable was level of education (time invariant). Persons with a lower level of education are less physically active then their higher educated counterparts [33]. Therefore, a five-point scale of the level of education analogous to that described by Lamprecht and colleagues [33] was used (Table 1). The third control variable was previous (in)activity duration (time varying). For each physically active episode or inactive episode, it was indicated (per year) how long this period had lasted already. Only the last 16 years were analyzed, but the data on LTPA over the entire life course were available. Therefore, the duration of the episode was already indicated before year 2004. This knowledge also helps to avoid the statistical problem of left-truncation or delayed entry [41], and the duration of staying in a particular episode is supposed to reduce the risk of change [21]. Due to the different ages of persons in the cohort, the maximal value of this variable was set to 15. The fourth control variable was quotient active years (time varying). In addition to taking into account the duration of the current episode, the number of physically active years in the life course can also affect LTPA [44]. For this purpose, the quotient of the respective physically active years and age was created to consider it. The fifth control variable was age group (time varying). LTPA also differs according to age [33]. Age did not have a linear effect for our study. Hence, age groups were formed based on the method of Wernli and Zella [45]: “1” (under 30 years of age); “2” (30–44 years); “3” (45–59 years); and “4” (60–76 years). To actively include the moderating effect of age and sex on a life event, interactions were calculated and reference categories were switched post hoc for precise statements on specific values [46].

Calculations were undertaken in R (R Foundation for Statistical Computing, Vienna, Austria) [47] with the package “lme4” [48]. The “glmer” command was used for this type of analysis [49] and if convergence issues occurred, the “bglmer” command from the package “blme” was referred to [50]. Thus, maximum likelihood estimations via numerical integration were suggested [41,51]. More precisely, the integral in GLMMs must be approximated and cannot be defined exactly. Therefore, the most reliable approximation is adaptive Gauss–Hermite–Quadrature [48], which was used here. When calculating models, variables were added stepwise and compared with the likelihood ratio test (chi-square test; *p* < 0.05) to identify the model with the best fit for the respective data [51], and the best-fitting model was reported.

## 3. Results

### 3.1. Descriptive Analyses

Taking up LTPA and especially terminating LTPA over 16 years did not occur very often (*n*_taking up LTPA_ = 514 (24% of the sample); *n*_terminating LTPA_ = 260 (12.1% of the sample)) (Table 1). This was underlined by the observation that 75.5% of the sample remained physically active (72.1%) and inactive (3.4%) for 16 years, in addition to the relatively long previous (in)activity duration, whereby these values were summed from the entire life course. The number of years of the previous activity duration (*n* = 24,209; mean = 13.12, SD = 3.92) compared with the number of years of previous inactivity duration (*n* = 3029; mean = 9.33, SD = 5.81) showed that activity episodes occurred more often and persisted for longer. The quotient active years were indicated along the same lines (mean = 0.64, SD = 0.21). When life events were regarded, a higher number of occupational life events was obvious, and there was a notably high number of job changes. Conversely, familial life events did not occur quite as often.

When looking at simultaneously occurring life events, it was observed that two life events in the same year occurred occasionally, but ≥3 life events were very rare. Considering the age group, we noted that more than three-quarters of the participants investigated were >45 years of age, which probably explained the frequency of certain life events to some extent. The level of education of this sample showed a slightly right-skewed distribution, but ≥15% of the study cohort was represented in the higher education level categories of 3 to 5.

### 3.2. Relationship between Life Events and Taking Up LTPA in the Life Course

Due to the physically active sample, the risk set for this calculation was rather small. All results presented below can be found in Table 2 (and all full models in Appendix A). Beginning with life events in the familial life domain, starting a relationship carried no association with taking up LTPA (logit = 0.64, *p* = 0.21). However, ending a relationship was positively related with taking up LTPA (logit = 1.31, *p* = 0.003). Thus, if a relationship broke up, the probability of taking up LTPA was significantly higher. Differences for sex or age group were not found (see RQ 1a). For becoming a parent, a reversed effect was detected: if this life event occurred, the probability was significantly lower for taking up LTPA (logit = −1.06, *p* = 0.02), which indirectly supports Hypothesis H1. Furthermore, differences between sexes or age groups did not emerge.

Job changes and starting vocational training as life events did not seem to be closely related with taking up LTPA (RQ 2a). Significant associations did not occur between taking up LTPA and starting vocational training (logit = 0.36, *p* = 0.17) or for starting a job (logit = 0.12, *p* = 0.63), and ending a job (logit = 0.17, *p* = 0.54), but retirement yielded a strong, positive effect on taking up LTPA (logit = 1.49, *p* < 0.001). Hence, persons of either sex who retired had a higher probability to take up LTPA. Thus, Hypothesis H2 was confirmed.

With regard to the number of simultaneously occurring life events (see RQ 3a), the model with age-group differences explained the data best. Considering the interpretation of results when interaction effects were modeled, persons aged 45–59 years (logit = 0.67, *p* = 0.02) and 60–70 years (logit = 1.20, *p* = 0.004) differed significantly from persons under 30 years of age (reference category; logit = –0.15, *p* = 0.43). To not only obtain the differences between age groups but to also compare people in an age group who had experienced life events with people of the same age group who had experienced no or fewer life events, the age group of interest was set as the reference category post hoc (not shown in Table 2, see Appendix A). Significant effects emerged for persons aged 45–59 years (logit = 0.52, *p* = 0.02) and 60–70 years (logit = 1.01, *p* = 0.004) but not for persons under 30 years of age (reference category in Table 2) and persons aged 30–44 years (logit = 0.06, *p* = 0.74). In summary, for persons between 45 years and 70 years of age, the number of simultaneously occurring life events was associated with a higher probability for taking up LTPA: the more life events, the higher the probability.

### 3.3. Relationship between Life Events and Terminating LTPA in the Life Course

For terminating LTPA, the risk set was much larger, and more person-years could be included in the calculation due to the physically active sample. All of these results are presented in Table 3 (and all full models in Appendix A). Looking at life events in the familial life domain, starting a relationship had no association with terminating LTPA (logit = 0.64, *p* = 0.16), but ending a relationship was linked to terminating LTPA (logit = 1.71, *p* = 0.003) and a significant interaction effect for sex was documented (logit = –2.52, *p* = 0.03) (see RQ 1b). The aforementioned main effect for ending a relationship showed the value for men who ended a relationship compared with men who stayed in a relationship, and it indicated a positive association with terminating LTPA, so men who ended a relationship were more likely to terminate LTPA. When the reference category was changed post hoc, a significant effect for women who ended a relationship did not emerge compared with women who stayed in a relationship for terminating LTPA (logit = –0.82, *p* = 0.43) (Appendix A).

For becoming a parent, the model with two interactions (becoming a parent × sex, and becoming a parent × age group) fitted the data best. For interpretation of these results, the values of the changed reference categories from Appendix A should be consulted directly. For men who became a parent, compared with men who do not attain fatherhood, no significant effects for terminating LTPA were detected for all age groups (<30 years: logit = −1.89, *p* = 0.13; 30–44 years: logit = –0.31, *p* = 0.77; 45–54 years: logit = 0.66, *p* = 0.65) (Appendix A). For women, the opposite was true. Almost in the entire life course, significant effects occurred for terminating LTPA when becoming a mother, compared with women who did not attain motherhood, with increasing effects with age (<30 years: logit = 1.26, *p* = 0.06; 30–44 years: logit = 2.84, *p* < 0.001; 45–54 years: logit = 3.76, *p* = 0.002) (Appendix A). In summary, the later in life one becomes a parent, the more likely women were to terminate LTPA, whereas men were not affected. Thus, Hypothesis H1 could be partially confirmed: for women, but not for men.

With regard to life events in the occupational life domain, the relationship between starting vocational training and terminating LTPA was dependent upon age (see RQ 2b). While there was no effect for people aged 15–29 years (logit = 0.17, *p* = 0.68), persons aged 30–44 years differed significantly (logit = 1.90, *p* = 0.004) and post hoc results from a changed reference category showed a higher probability for terminating LTPA within this age range (logit = 2.07, *p* < 0.001) (Appendix A). Starting a job did not seem to affect termination of LTPA (logit = 0.02, *p* = 0.93), whereas ending a job was positively associated with terminating LTPA in general (logit = 0.82, *p* < 0.001) with no sex or age group differences (see RQ 2b). The final ending of a job (retirement) was not related to termination of LTPA (logit = –0.36, *p* = 0.62).

If multiple life events occurred simultaneously, significant differences between age groups emerged (see RQ 3b). Post hoc results from the changed reference categories (Appendix A) yielded significant effects for terminating LTPA upon experiencing multiple life events for persons aged 15–29 years (logit = 0.30, *p* = 0.04) and even more for those aged 30–44 years (logit = 0.83, *p* = 0.01). If older than 44 years, multiple life events did not affect termination of LTPA (45–59 years: logit = 0.16, *p* = 0.48; 60–72 years: logit = 0.40, *p* = 0.38).

## 4. Discussion

In this study, transitions into and out of LTPA in the life course were investigated when experiencing single and multiple life events in the familial and occupational life domains. Using retrospective longitudinal data over 16 years from persons aged 16–76 years, each age in the life course was covered. In addition to the known effects of single life events on LTPA in “common” age ranges [10], age differences and the occurrence of multiple simultaneous life events were investigated for the first time. Overall, life events were more closely associated with terminating LTPA than with taking up LTPA. However, these data should not be viewed only negatively, as they can also be seen as an opportunity at certain life stages. Thus, the timing of events [5,16,19] is crucial for certain life events. Furthermore, the interdependencies between life domains were therefore partially evident and could be an indication that the occurrence of (multiple) life events competed with the resources between the respective life domains. However, they could also be interrelated in a supportive, complementary, and compensatory way [17,27,28].

This results provide new insights into how life events are related with LTPA over the life course and are partly in accordance with data from previous studies. A summary of the results is shown in Table 4. From the familial life domain, starting a relationship did not have an association with transitions in LTPA in the life course (see RQ 1) and, thus, showed neither positive nor negative effects [10]. In contrast, ending a relationship in the life course increased the likelihood of taking up LTPA, but there was also a higher probability of terminating LTPA for men. Hence, women tended to have or invested more resources for LTPA after a break-up (perhaps in the sense of a compensation effect), whereas men seemed to have a change in their resources that could lead to taking up or terminating LTPA.

Becoming a parent is, in general, not conducive to LTPA [13,24], but interesting differences between sexes and age were shown in this study. For both sexes and over the entire life course, taking up LTPA was less likely if you became a parent, but terminating LTPA differed between sexes (see H1) and age groups: Men were not affected, while for women of increasing age, an even higher risk for terminating LTPA was documented. These observations could be explained by the additional resources needed for mothers than for fathers (life domains compete for resources). In addition, the later the birth in the life course, the more the person–environment fit that had been accustomed over time could be disrupted, and adjustment seemed to be more difficult [25].

With regard to life events in the occupational life domain (see RQ 2), starting vocational training did not yield the suggested negative effect regarding LTPA in youth and early adulthood [10,14]. This finding may have been due to a different education system in Switzerland or due to compensating or supporting resources between the life domains (e.g., recovery from educational stress by using LTPA). However, we did not examine specific vocational training (e.g., entering university) but instead general vocational training with a state-recognized certificate, and only vocational training from the age of 15 years was included (where transition to secondary schools takes place in Switzerland). Nevertheless, starting vocational training at 15–29 years of age was not related to terminating or taking up LTPA. Surprisingly, starting vocational training after 29 years of age was associated with terminating LTPA. The timing of life events is crucial, as demonstrated here by the start of vocational training in a rather “uncommon” phase of life. One possible reason could be competing resources between the life domains (e.g., the resource time due to already consolidated life circumstances). In the present study, when starting a job, the postulated negative effect on LTPA [10,14] was not found for young adulthood (possibly because of a different education system in Switzerland) or for the entire life course. We did not separately investigate the first entry into the labor market, where the largest effects occur [10]. However, starting a job in the life course did not seem to alter individual resources to change LTPA behavior, or it could be explained by compensating or supporting effects (e.g., maintaining a work–life balance due to LTPA). The reported inconclusive effects of ending a job on LTPA [10] suggest in the present study that this life event leads only to a higher probability for terminating LTPA but has no effect on taking up LTPA. It seems that the possibly newly gained resources by ending a job are not used for LTPA or that compensation of occupational strains through LTPA is no longer necessary. However, whether a new job or training was started immediately afterwards was not investigated. The postulated positive effect of retirement on LTPA^31^ was shown with the high probability of taking up LTPA upon retirement (see H2). The newly gained resources could explain this association.

Simultaneously occurring life events showed the redistribution of resources in both ways and with age differences (see RQ 3). The more life events experienced at 45–70 years of age, the higher the probability of taking up LTPA, whereas this relationship was not found for persons aged 15–44 years. Thus, in older adulthood, the “window of change” due to multiple life events was used in a positive way with regard to LTPA. In younger ages (15–44 years), the more life events experienced, the more likely was termination of LTPA, whereas this effect was not present for older persons (45–70 years). Therefore, in this case, the window of opportunity led to a negative change concerning physical activity, as shown by Oman and colleagues but for older adults (>50 years) [32]. Hence, until 44 years of age, multiple life events were related to LTPA termination whereas, after 44 years of age, they were linked to an increased chance of taking up LTPA. The explanation of a larger adjustment when more life events occurs seems plausible, but the direction of the change appears to depend on age, and the reasons for the direction of change are not clear.

The timing of events is interesting because age differences for becoming a parent and starting vocational training on terminating LTPA indicate an effect at mainly non-normative life stages [26] for these life events (becoming a parent until 54 years; starting vocational training at 30–44 years of age). Moreover, timing seems to be the crucial part when considering simultaneously occurring life events and LTPA, with different or even opposite effects depending on age. Differences between sexes occurred only for familial life events. Therefore, these life events seemed to be affected by sex-specific behavior and possibly societal expectancies. Furthermore, besides the investigated moderating effect of sex and age group, the effect of the education level together with life events was tested on LTPA. Tendencies (*p* < 0.10) for just two life events were noted. The already reported relationship between ending a job and terminating LTPA could be “cushioned” by higher education, and the latter decreased the probability of terminating LTPA even more when someone retired. Consequently, a higher level of education could be seen as another type of individual resource for the occupational life domain.

We were able to show novel results by: (i) investigating concurrent taking up and terminating LTPA in the life course, which also uncovered competing effects (e.g., for ending a relationship in men); (ii) differentiating for age groups (i.e., certain life events depend on age when regarding the effect on LTPA); (iii) showing the reasoned effect of simultaneously occurring life events on LTPA with a higher redistribution of resources, which was revealed to be different for the age groups we evaluated.

### 4.1. Limitations

Some limitations have to be considered for this study. First, LTPA was self-reported and retrospective. Knowing that this is not the most valid method [52], as it often yields an overestimate of physical activity [53], efforts were made to gather the most reliable and valid data possible. To do this, we used a reliability test, although a lack of validity cannot be ruled out. Furthermore, transitions regarding LTPA (terminating or taking up) were investigated but not decreases or increases in LTPA. Therefore, the biggest change was measured, but strong decreases or increases were not captured. In addition, to be considered physically active in this study, the minimum of once per week was used. However, we have no information on intensity and frequency (except at least once a week), which is poorly captured in retrospective studies in general [40]. Therefore, it is not possible to say with certainty whether the physical activity recommendations are met. Moreover, in the physically active sample, the risk set for taking up LTPA was small. This led to relatively low statistical power for calculations with life events and taking up LTPA. Thus, the absence of significant interaction effects could perhaps be attributed to (among others) low statistical power. Additionally, many life changes occur in the first chosen age group up to 29. This age group was not further differentiated, in addition to the reasons given in Section 2.4, due to the age distribution of the sample. As shown in Table 1, this age group is still the smallest. Beyond that, the duration of an active episode or inactive episode after a transition (taking up or terminating LTPA) was not taken into account but was set to ≥1 year. Hence, it is not known if, for instance, after terminating LTPA, the subsequent inactive episode lasted just 1 year or 7 years. Moreover, the interrelations between the life domains when regarding resources were not shown and, thus, could not be uncovered empirically. Lastly, this study covers life events in the familial and occupational life domains, which are highly relevant to this topic [10]. Other relevant life events, such as those that are health related or residence related, are not included here.

### 4.2. Future Research

New findings have been gained regarding age differences and the number of life events experienced. Consequently, when examining life events and changes in LTPA, age and the possibility of multiple life events occurring simultaneously must be considered for future research. In addition, the general theoretical approach of the life course cube with the spillover effect and related changing resources was applied to LTPA and life events for the first time. Therefore, further research should examine in further depth how this mechanism of the spillover effect with changing resources works in specific constellations (different and combined life events in various life domains and different age groups) in order to derive precise implications and more specific theories. Furthermore, our study should be replicated prospectively and changes integrated in the LTPA level with objective measurement methods.

## 5. Conclusions

The positive and/or negative associations of life events on taking up or terminating LTPA can occur identically over the life course or be limited to specific life stages, depending on the type and number of life events. Therefore, for inactive persons, supporting programs or interventions can be considered when experiencing (multiple) life events for a positive change regarding LTPA. Opportunities for cushioning the possible adverse effects of life events for LTPA should be developed for active persons. One promising approach (indicated by the results for control variables in the Appendix A) is the (early) promotion of LTPA in general. This is because the more physically active years a person has had in their life, the higher the chance is of taking up LTPA again. Moreover, the longer the previous active episodes lasted, the lower the risk of them terminating LTPA.

## Figures and Tables

**Table 1 ijerph-18-09809-t001:** Descriptive data for the entire sample *n* = 1857 persons over 16 years (total *n* = 27,238 person-years).

Variable	Number of Events	% of Persons in the Sample	
*Dependent variables*			
Taking up LTPA	514	24%	
Terminating LTPA	260	12.1%	
*Independent variables*			
Familial life domain			
Starting a relationship (12–72 years)	339	16%	
Ending a relationship (12–72 years)	281	14%	
Becoming a parent (15–54 years)	418	13.8%	
Occupational life domain			
Starting vocational training (15–44 years)	355	21.2%	
Starting a job (15–70 years)	1324	42.9%	
Ending a job (15–70 years)	1537	40.2%	
Retirement (50–72 years)	552	29.7%	
Simultaneously occurring life events (15–70 years)			
1	2623	73%	
2	913	32.5%	
3+	165	7.2%	
**Variable**	** *n* ** **(%)**	**Mean**	** *SD* **
*Control variables*			
*Time invariant:* Sex			
Male (0)	738 (39.7%)		
Female (1)	1119 (60.3%)		
*Time invariant:* Level of education (1–5)			
Compulsory school (1)	183 (9.9%)		
Secondary school/lower professional education (2)	758 (40.8%)		
Higher professional education leaving certificate (3)	329 (17.7%)		
Technical college (4)	302 (16.3%)		
University (5)	285 (15.3%)		
*Time varying:* Previous (in)activity duration (1–15) ^1^			
Previous activity duration	24,209	13.12	3.92
Previous inactivity duration	3029	9.33	5.81
*Time varying:* Quotient active years ^1^	27,238	0.64	0.21
*Time varying:* Age group ^1^			
until 29 (ref.)	4311 (15.83%)		
30–44	4939 (18.13%		
45–59	10,903 (40.03%)		
60–76	7085 (26.01%)		
*Additional information about the sample*			
Always physically active (over 16 years)	1338 (72.1%)		
Always physically inactive (over 16 years)	63 (3.4%)		
At the time of the survey physically active	1741 (93.8%)		

Note: SD = standard deviation. ^1^ Values in person-year due to time-varying control variables.

**Table 2 ijerph-18-09809-t002:** Multilevel discrete-time event history analysis with separate calculations for each life event and taking up leisure-time physical activity, controlled for sex, level of education, previous inactivity duration, quotient active years, and age group (for each complete model reported, see Appendix A).

Life Event	Taking Up LTPA			
	*n*	Logit	*SE*	*P*
*Familial life domain*				
Starting a relationship (12–72 years) ^1,2^	804	0.64	0.51	0.21
Ending a relationship (12–72 years) ^1^	1586	1.31	0.44	0.003 *
Becoming a parent (15–54 years) ^1^	1391	−1.06	0.45	0.02 *
*Occupational life domain*				
Starting vocational training (15–44 years) ^1,2^	759	−0.05	0.46	0.91
Starting a job (15–70 years) ^1,2^	2025	0.12	0.25	0.63
Ending a job (15–70 years) ^1,2^	1995	0.17	0.27	0.54
Retirement (50–72 years) ^1^	935	1.49	0.44	<0.001 *
*Simultaneously occurring life events (15*–*70 years*)				
Simul. occurring life events × age group (ref. = 1; <30 y)	2149	−0.15	0.19	0.43
age group 2 (30–44 y)		0.21	0.26	0.42
age group 3 (45–59 y)		0.67	0.29	0.02 *
age group 4 (60–70 y)		1.20	0.42	0.004 *

Note: SD = SE = standard error; * *p* < 0.05. ^1^ No significant age group differences or sex differences were found. ^2^ Model with just control variables and without life event fits best, but to show the non-significant effect of life event, this model is presented.

**Table 3 ijerph-18-09809-t003:** Multilevel discrete-time event history analysis with separate calculations for each life event and terminating leisure-time physical activity, controlled for sex, level of education, previous activity duration, quotient active years, and age group (for each complete model, see Appendix A).

Life Event	Terminating LTPA			
	*n*	Logit	*SE*	*P*
*Familial life domain*				
Starting a relationship (12–72 years) ^1,2^	5427	0.64	0.46	0.16
Ending a relationship (12–72 years)				
Ending a relationship × sex (ref. = men)	17,741	1.71	0.57	0.003 *
sex (women)		–2.52	1.17	0.03 *
Becoming a parent (15–54 years) ^3^				
Becoming a parent × age group (ref. = 1; <30 y) × sex (ref. = men)	12,689	−1.89	1.25	0.13
age group 2 (30–44 y)		1.57	0.72	0.03 *
age group 3 (45–54 y)		2.50	1.34	0.06
sex (women)		3.15	1.09	0.004 *
*Occupational life domain*				
Starting vocational training (15–44 years)				
Starting voc. training × age group (ref. = 1; <30 y)	5385	0.17	0.43	0.68
age group 2 (30–44 y)		1.90	0.66	0.004 *
Starting a job (15–70 years) ^1,2^	19,690	0.02	0.26	0.93
Ending a job (15–70 years) ^1^	19,355	0.82	0.22	<0.001 *
Retirement (50–72 years) ^1^	10,653	–0.36	0.74	0.62
*Simultaneously occurring life events (15*–*70 years*)				
Simul. occurring life events × age group (ref. = 1; <30 y)	21,783	0.30	0.15	0.04 *
age group 2 (30–44 y)		0.53	0.21	0.01 *
age group 3 (45–59 y)		–0.13	0.28	0.63
age group 4 (60–70 y)		0.09	0.47	0.84

Note: SD = SE = standard error; * *p* < 0.05. ^1^ No significant age group differences or sex differences were found. ^2^ Model with just control variables and without life event fits best, but to show the non-significant effect of life event, this model is presented. ^3^ The model with two 2-way interactions (becoming a parent × age group and Becoming a parent × sex) fits best and is therefore presented.

**Table 4 ijerph-18-09809-t004:** Summary of results for the relationship between life events and taking up or terminating LTPA.

Life Event	Taking Up LTPA	Terminating LTPA
*Familial life domain*		
Starting a relationship (12–72 years; RQ 1)	*No relationship*	*No relationship*
Ending a relationship (12–72 years; RQ 1)	↑	↑ For men
Becoming a parent (15–54 years; H 1)	↓	↑ For women, with an increasing effect with age
*Occupational life domain*		
Starting a vocational training (15–44 years; RQ 2)	*No relationship*	↑ For 30–44 years old
Starting a job (15–70 years; RQ 2)	*No relationship*	*No relationship*
Ending a job (15–70 years; RQ 2)	*No relationship*	↑
Retirement (50–72 years; H2)	↑	*No relationship*
Simultaneously occurring life events (15–70 years; RQ 3)	↑ For 45–70 years old	↑ For 15–44 years old

Note: ↑ means a significant positive statistical relationship; ↓ means a significant negative statistical relationship.

## Data Availability

Publicly available datasets were analyzed in this study. These data will be provided upon acceptance of the manuscript.

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
