# Peer review of "Taking Up and Terminating Leisure-Time Physical Activity over the Life Course: The Role of Life Events in the Familial and Occupational Life Domains"

_ijerph, 2021, doi:10.3390/ijerph18189809_

Round 1

Reviewer 1 Report

I am glad most of the suggestions have been incorporated into the text, not all though but that is fine with me. 

Reviewer 2 Report

The authors have revised and improved their paper considerably and addressed almost all of my initial concerns. Thank you! Hence, I can now recommend the acceptance of the paper for publication.

Reviewer 3 Report

The paper is very complete after the modifications made.
In this version, the methodological aspects of the research carried out can be seen more clearly.
The survival analysis used is adequate and useful for the objectives of this research.
The conclusions are correct and concrete, in accordance with the results obtained.
Perhaps the specification in this article of the total number of items of the above-mentioned survey (sport schweiz 2020) that have been considered for this study would be desirable.

This manuscript is a resubmission of an earlier submission. The following is a list of the peer review reports and author responses from that submission.

Round 1

Reviewer 1 Report

This is an interesting paper that addresses an important question about how work and life experiences influence leisure-time activity. The study authors use a novel approach to answering their question from a life course perspective. I was unable to access the supplementary material which could provide some answers to my concerns below.

A few minor changes are noted below. A more significant concern is the limited detail about the methods of data collection. What was the timing of data collection? Currently, my interpretation from the methods described is that data collection was in 2019 and retrospectively examined 16 years. If so, this is a major concern. The study was described as longitudinal in another place and, if such, I would expect that the same data were collected repeatedly over multiple years. Please clarify this in the methods.

An additional concern was regarding the measurement of physical activity. I would like to see more detail in the paper about the measure that was used to assess this construct. The current description seems very limited and appears to allow for very low level of physical activity (once per week). Additionally, it doesn't appear that the intensity of activity is measured and this should be included in the limitations. Further, without description of the intensity or frequency, readers are unable to determine if participants are meeting recommendations for physical activity. The estimation that 70+% of participants were continuously active doesn't align with population data on physical activity.

Introduction

Page 1, Line 31 - replace the word "tool" with behavior

Page 2, Line 61 - replace the word "activity" with behavior

Page 2, Line 78 - clarify what is meant by  "compensate respectively substitute mutually"

Page 2, Line 88 - delete the word "rather"

Page 3, Line 108 - occurred or occurring?

Results

Page 8, Line 290 - change the "fitted" to

This paper offers some insight for life experiences that influence LTPA but there are a number of limitations to the study approach as currently described that prevent any significant conclusions from being drawn from this study. However, this is an important call to action for more studies to examine these questions.

Reviewer 2 Report

The paper is well focused and reaches interesting conclusions that can contribute to healthier lifestyle planning.

The methodology and design are correct for the type of study presented.

Obviously there are some limitations when considering external validity, in the sense of extrapolating these behaviours to populations other than Switzerland. But nevertheless, the study is interesting.

Regarding the test-test, in my opinion they could have used a larger sample, which would have given more certainty when assessing the significance of the Reliability Coefficient.

As a suggestion, I would like to point out that in my environment, researchers tend to interpret the Odds Ratio (the exponential of the "logits") more easily.

Therefore, it might be useful to add it to make interpretation a little easier.

I congratulate you on your work.

Reviewer 3 Report

The authors conducted an interesting and original study that contributes to the state of knowledge by showing how certain life events influence the level of LTPA. The paper is based on a thorough data analysis and a clever research design and comes to meaningful results and conclusions. However, there are also some weaknesses: The theory section and the development of research questions is a bit muddled and needs better structure. Some of the procedures and measures must be explained with more clarity and in greater depth. Overall, I think these flaws can be addressed in a major revision of the paper and I encourage the authors to prepare a revised manuscript.

  • Page 1: Please first summarize the state of research and then point to some research gaps. This paper immediately starts with research gaps. I suggest to giving some credit to existing studies at first so that readers can better evaluate what sort of research is still needed.
  • Page 2: Please better explain the key assumptions and key strengths of a life-course approach. The “life course cube” might be a nice model but it remains underspecified and cannot replace a proper theoretical foundation. For instance, there is only a very loose connection between life events, “resources” and LTPA.
  • Page 2: Give readers more context when you refer to previous studies. Please mention key aspects like methods, results, geographic information that help readers to understand how reliable and transferrable a study finding is.     
  • Page 2: Please be clearer in your hypothesis and assumptions. Sometimes you refer to “research questions”, sometimes to “hypotheses”, which is confusing. Either way, however, you should be more specific to justify why you expect a certain effect or why the research question is important.
  • Page 3: Sampling procedures are a bit strange and not well explained. Why did you combine two ways of recruiting participants (Statistical Office + Survey Institute)? How does this affect the representativeness of the sample?
  • Page 3: Please explain the measurement of LTPA better, incl. the wording of the question. I can understand that people can remember well when they started or terminated “sport” – in a narrower sense – but you claim to measure physical activity. So what is included in LTPA? What about daily cycling or walking activities? How did you explain to the interviewees what starting/terminating these activities means?
  • Table 1 and 2: Why are the effects of the control variables not reported?
  • Page 9: First sentence of the discussion states that “health-relevant transitions” were measured. That is just a claim. You measured “transitions”, if these transitions have any impact on health was not investigated. I suggest refraining from these claims.
  • Page 10: Simultaneous life events affect LTPA differently in age groups. Can you try to figure out which life events are typically combined in the age groups you analyze? How do these combinations typically affect ressources (time, money etc.)? The paper remains quite vague with pointing to resources.
  • Page 11: Another limitation is that only events from two life domains were investigated. But there may be a variety of other (critical) life events that impact on LTPA (e.g., getting a chronic illness, relocation).

Reviewer 4 Report

Thank you for a chance of reviewing this interesting study. 

I believe this study has some academic value and with a little bit of enhancement will make a good scholarly publication. 

Abstract is informative and concise. I suggest considering adding 'family' to the key words. 

Introduction is interesting and it contains theoretical background, which is valid in this kind of studies. All abbraviations are explained which is also good for the potential readers. I suggest expanding a little bit on the rationale for the study by adding some more works on 'health-related fitness links between parents and their child'. and looking into "Tracking of Physical Activity across the Lifespan" this paper may also help. 

Methods 

The sample size is impressive and duration of the study as well - the method of recruiting the participants is well-desribe and reasonable.

If the authors are in possession of more co-efficent data confiriming validity and reliability of the research instruments please provide other figures in the text as well. 

Division of the age-groups seems fine, however, the First group 'under 30'

seems to be quite broad and includes 15 as well as 30 years old people - this age difference might be an issue as participants from this group might be at different moments of their life course. 

In Results, which are well-described I found two tables numbered 1! Please check numbering. 

Discussion is solid, it links authors' own findings with those of other authors. I am not sure it table 3 with results should belong to this section or should be moved to the Results, even though it is summarizing, but I leave it up to the authors to decide. 

And also, since in conclusions the authors refer to early promotion of LTPA, I think in Discusison you should more expand and explore links with social support (look for example in "the analysis of social support level in foster familites in the context of thier leisure time activities" or  look at some intervention in the family setting in early years increasing chances for regular PA "Family leisure-time physical activities - results of the juniors for seniors 15-week intervention programme' or 'Associations between the home yard and preschoolers’ outdoor play and physical activity'. 

Tables are neat, but numbering requires attention.   
